# Impact of anthropometry training and feasibility of 3D imaging on anthropometry data quality among children under five years in a postmortem setting

Priya M. Gupta[1], Kasthuri Sivalogan[1], Richard Oliech[2], Eugene Alexander[3], Jamie Klein[4], O. Yaw. Addo[1,6], Dickson Gethi[5], Victor Akelo[5], Dianna M. Blau[6], Parminder S. Suchdev[1,4,6¤]*

1 Nutrition and Health Sciences Program, Laney Graduate School, Emory University, Atlanta, Georgia, United States of America, 2 Kenya Medical Research Institute, Kisumu, Kenya, 3 Body Surface Translations, Inc., Athens, Georgia, United States of America, 4 Department of Pediatrics, Emory University School of Medicine, Atlanta, Georgia, United States of America, 5 US Centers for Disease Control and Prevention-Kenya, Kisumu and Nairobi, Kenya, 6 US Centers for Disease Control and Prevention, Atlanta, Georgia, United States of America

¤ Current Address: Department of Pediatrics, Emory University, Atlanta, Georgia, United States of America
* psuchde@emory.edu

**Data Availability Statement:** The minimal anonymized data set necessary to replicate our

## Abstract

### Background

The Child Health and Mortality Prevention Surveillance Network (CHAMPS) identifies causes of under-5 mortality in high mortality countries.

### Objective

To address challenges in postmortem nutritional assessment, we evaluated the impact of anthropometry training and the feasibility of 3D imaging on data quality within the CHAMPS Kenya site.

### Design

Staff were trained using World Health Organization (WHO)-recommended manual anthropometry equipment and novel 3D imaging methods to collect postmortem measurements. Following training, 76 deceased children were measured in duplicate and were compared to measurements of 75 pre-training deceased children. Outcomes included measures of data quality (standard deviations of anthropometric indices and digit preference scores (DPS)), precision (absolute and relative technical errors of measurement, TEMs or rTEMs), and accuracy (Bland-Altman plots). WHO growth standards were used to produce anthropometric indices. Post-training surveys and in-depth interviews collected qualitative feedback on measurer experience with performing manual anthropometry and ease of using 3D imaging software.

study findings has been included as Supporting information.

**Funding:** Sources of Support: This work was funded by grant OPP1126780 from the Bill & Melinda Gates Foundation. The funder participated in discussions of study design and data collection. They did not participate in the conduct of the study; the management, analysis, or interpretation of the data; preparation, review, or approval of the manuscript; or decision to submit the manuscript for publication.

**Competing interests:** I have read the journal's policy and the authors of this manuscript have the following competing interests: Eugene Alexander holds an ownership position in Body Surface Translations and therefore has a financial interest in the success of the 3D testing device described in this study. Data were blinded and not shared with Mr. Alexander until completion of draft manuscript. This does not alter our adherence to PLOS ONE policies on sharing data and materials. Additional disclosure: The findings and conclusions in this report are those of the authors and do not necessarily represent the official position of the Centers for Disease Control and Prevention.

## Results

Manual anthropometry data quality improved after training, as indicated by DPS. Standard deviations of anthropometric indices exceeded limits for high data quality when using the WHO growth standards. Reliability of measurements post-training was high as indicated by rTEMs below 1.5%. 3D imaging was highly correlated with manual measurements; however, on average 3D scans overestimated length and head circumference by 1.61 cm and 2.27 cm, respectively. Site staff preferred manual anthropometry to 3D imaging, as the imaging technology required adequate lighting and additional considerations when performing the measurements.

## Conclusions

Manual anthropometry was feasible and reliable postmortem in the presence of rigor mortis. 3D imaging may be an accurate alternative to manual anthropometry, but technology adjustments are needed to ensure accuracy and usability.

## Introduction

Malnutrition is estimated to contribute to approximately half of under-5-mortality (U5M) [1–3]. Malnutrition is also a major cause of morbidity as malnutrition plays a critical role in child neurodevelopment and health across the life course [2–4]. Reliable assessment tools for malnutrition are essential to reflect individual status, measure biological function, and predict health outcomes [5–7]. In children, inadequate growth is defined according to anthropometric measurements (length, weight, head and mid-upper arm circumference) that fall below 2 standard deviations of the normal sex-specific weight-for-length (wasting), length-for-age (stunting), and weight-for-age (underweight) [7]. Despite the importance of accurate anthropometry to detect early signs of malnutrition and monitor child growth, health facilities routinely use non-standardized anthropometric equipment, and as a result, measurements are often inaccurate [8]. Inaccurate measurements can lead to spurious classification of malnutrition in both individuals and populations [9].

In addition to the challenges of procuring and using standard anthropometric measurement tools, anthropometric measurements are subject to human error and are particularly difficult to collect among young children as children are easily distressed, have difficulty staying still, and may be unable to meet the requirements (i.e. ability to lie down or stand up) for manual anthropometry [10–12]. Anthropometric measurements are particularly challenging in hospitalized settings or in medically complex patients due to medical equipment that may impede taking measurements (e.g., intravenous lines or feeding tubes), severe illness, or limitations in mobility. These children are also at highest risk of malnutrition [8, 13]. Additionally, qualitative findings from a quality improvement study in a children's hospital found that, wooden height-length measuring boards (ShorrBoard®, Weigh and Measure, LLC, Maryland USA) were considered to be "*heavy, cumbersome to assemble, frightening to patients, and required pre-planning and coordination between clinical staff with busy schedules and competing priorities*" [8]. Lastly, in field settings, the weight of the board may impede transportation and movement within the field and lack of standardization and maintenance of anthropometric equipment across study sites may contribute to poor data quality and misclassification [10, 11]. The post-mortem setting is another environment in which manual anthropometry may be challenging. Morgue capacity, rigor mortis, and edema can impact the quality and accuracy of

measurements [14]. To our knowledge, no research has been conducted on the feasibility of using gold-standard anthropometric assessment in the postmortem setting.

The Child Health and Mortality Prevention Surveillance (CHAMPS) network is a multi-site surveillance system which strives to identify and understand the causes of under-5-mortality (U5M) in seven surveillance sites in sub-Saharan Africa and South Asia through detailed cause of death attribution with the use of high-quality postmortem anthropometrics, tissue samples, clinical abstraction, verbal autopsy, and the ability to integrate data from site-specific health and demographic surveillance systems (HDSS) [15, 16]. A recent analysis of the postmortem anthropometric data in CHAMPS suggested that nearly 90% of cases 1–59 months had evidence of undernutrition (stunting, wasting, or underweight) [17]. Given these data, it is possible that malnutrition is directly or indirectly associated with child mortality. However, our understanding of the relationship between malnutrition and mortality may also be hindered by poor anthropometric measurement data quality, including digit preference (e.g. measurement rounding), high percentage of biologically implausible values, and standard deviations for anthropometric indices that exceed acceptable limits, which may lead to misclassification of malnutrition [18–20]. These data quality and precision outcomes may be a result of shortages of standard equipment in CHAMPS sites, lack of training on manual anthropometry, or difficulty in conducting manual anthropometry in the postmortem setting (rigor mortis, poor lighting in morgue facilities).

Our primary objectives were to determine whether manual anthropometry is feasible in the postmortem setting and to quantify the impact of training and standard equipment on data quality. Given the practical challenges of performing manual anthropometry in field and hospital-based settings, various 3D imaging approaches have also been developed to obtain anthropometric measurements. An efficacy study conducted at Emory University found that a 3D imaging software was as accurate as gold-standard manual anthropometry among under-5 children in Atlanta-area daycare centers [10]. However, data are also needed to assess 3D imaging in challenging hospital- or field-based settings. Therefore, our secondary objective was to assess the validity and acceptability of 3D imaging for anthropometric assessment compared to gold-standard manual anthropometry.

## Materials and methods

### Study site and data collection

This longitudinal quality improvement study adopted a mixed-methods approach utilizing quantitative and qualitative research on the experience conducting manual anthropometry and 3D imaging in the postmortem setting. The study took place from October 2018 to September 2019 in the CHAMPS Manyatta, Kenya site located at the Jaramogi Oginga Odinga Teaching and Referral Hospital (JOOTRH). Prior to the training, site staff performed manual anthropometry on 75 deceased children as a routine part of the minimally invasive tissue sampling (MITS) portion of CHAMPS data collection. The MITS procedure is an abridged postmortem examination technique that has been validated for cause of death investigation in low-resource settings, described in detail in an earlier study [21]. Written informed consent was obtained from families as part of the CHAMPS enrollment procedures. The CHAMPS protocol was approved by ethics committees in Kenya and at Emory University, Atlanta, GA, USA. Additional information regarding the ethical, cultural, and scientific considerations specific to inclusivity in global research is included in the Supporting Information.

Upon conclusion of pre-training data collection, a senior nutritionist, pediatrician, and anthropometry expert led and conducted an on-site 1-week training on manual anthropometry and the 3D imaging scanner for 6 staff. Using materials developed by the CDC, WHO and

UNICEF, the training on manual anthropometry emphasized best practices for accurate manual measures of length, weight, head circumference (HC) and mid-upper arm circumference (MUAC) measurements using two trained anthropometrists and standard operating procedures [22]. Standard equipment in both sites, including wooden height-length measuring boards (ShorrBoard®, Weigh and Measure, LLC, Maryland USA), digital scales (Rice Lake Weighing Systems, Inc., Rice Lake, WI), and standard tape measures (Weigh and Measure LLC, Maryland USA), were used to ensure accurate measurement of recumbent length, weight, HC and MUAC, respectively. Staff completed an anthropometry standardization exercise using live children to ensure competence in conducting manual anthropometry. Staff were also trained on proper use the 3D imaging software using dolls and live children; details on the imaging software are provided in earlier studies [10, 23, 24]. Briefly, the AutoAnthro system uses an iPad™ tablet, and a Structure Sensor™ camera attached to the tablet to capture non-personally identifiable anthropometric scan images of the deceased child. Following the training, two trained anthropometrists manually collected anthropometric measurements for 76 cases, with two separate measurements collected per case by different anthropometrists. Additionally, 3D scans were completed in duplicate for each anthropometrist, for a total of 4 scans per case. During data processing, after the completion of data collection, it was identified that the AutoAnthro software settings had been inadvertently altered for a significant number of cases, resulting in a final sample size of 23 cases.

## Outcomes of interest

Key outcomes of interest included measures of data quality, precision, and accuracy. Data quality outcomes indicators included digit preference and standard deviations (SD) of anthropometric indices. Digit preference is the examination of a uniform distribution of terminal digits. We also calculated a digit preference score (DPS) to evaluate digit preference [25]. The DPS ranges from 0 to 100. Scores are low in instances of high agreement with the ideal of non-preference of the terminal digits, whereas DPS rises as the measures deviate from a uniform distribution across the terminal digits 0 through 9. In previous studies, a DPS cutoff above 20 was used to define the presence of digit preference [10, 26]. We thus used DPS<20 as acceptable, and DPS≥20 to indicate digit preference was problematic. Previous studies have suggested acceptable standard deviation ranges specifically for data quality among living children [27]. These include 1.10–1.30, 1.00–1.20, 0.85–1.10 for length-for-age (HAZ), weight-for-age (WAZ), and weight-for-length (WLZ) z-scores, respectively. Z-scores for anthropometric indices were produced using the WHO Multicentre Growth Reference Study anthro R package [28].

Technical errors of measurement (TEM) were used to assess measurement precision. Following the training, the site staff performed manual anthropometry in duplicate. It is important to note that this differs from the data collection strategy pre-training in which a single set of measures were taken. As a result, we were only able to calculate TEMs for the data post-training in both sites. TEM express the error margin in anthropometry; they are unitless and allow comparison of errors across measures (e.g., weight, height etc.). Absolute TEMs were calculated using the formula outlined in Equation 1 (Table 4). Absolute TEMs can also be transformed into relative TEMs, which express the error as a percentage corresponding to the total average. Relative TEMs (rTEM) were calculated using the formula outlined in Equation 2 (Table 4). We used a cutoff of <1.5% rTEM to indicate a skillful anthropometrist [25].

Finally, Bland Altman plots were used to assess the accuracy of the 3D imaging software relative to manual anthropometry following the training and were quantified in the unit of the measure (cm or kg). Spearman correlation coefficients examined the strength of the relationship between scans and manual measures.

Following the study, a short survey was sent to the 6 study participants. The survey collected information on whether the participants believed training on manual anthropometry improved the accuracy of the measurements, whether 3D imaging reduced the time to measure, and asked about the participants preference in measuring using manual anthropometry or the 3D imaging technology. We also conducted a 60-minute in-depth interview with the single lead site technician to collect qualitative feedback on the team's experience with performing manual anthropometry and ease of using the 3D imaging software. All analyses were conducted in R statistical software [29]. Statistical tests were two-sided and evaluated using an alpha level equal to 0.05. Pearson's Chi-Square tests (categorical variables) or t-tests (continuous variables) were used to evaluate differences between pre-intervention and post-interventions groups. The qualitative data were analyzed using simple frequencies and applying manual thematic analysis; findings informed the implementation of manual anthropometric measurements across the CHAMPS Network.

We also conducted a small study in collaboration with the Pediatrics and Pathology departments at Children's Healthcare of Atlanta, Egleston Hospital (CHOA). The goal was to evaluate whether manual anthropometry and 3D imaging performed consistently in a high-resource setting with adequate lighting and internet. The same training, detailed above, was used, and pathology staff notified the anthropometrists upon arrival of a case at the morgue. Manual anthropometry was to be performed prior to the start of the diagnostic autopsy. Significant challenges arose during data collection, including identification of eligible cases and timing to conduct anthropometry before the start of the diagnostic autopsy. Despite best efforts to coordinate between the study team and CHOA team, the study resulted in a limited sample size of 3 cases; thus, our results will focus on the Kenya site.

## Results

Sample characteristics are summarized in Table 1. Most children were under 2 years of age and were evenly distributed by sex. There were no significant differences in demographic characteristics or anthropometric measurements between the pre- and post- training groups. The prevalence of stunting, wasting, and underweight were overall high, with a higher prevalence of stunting noted in the post-training group (p = 0.02).

### Evaluation of quality- digit preference

In Table 2, prior to training, there was a clear tendency to round to the nearest 0.0 or 0.5 decimals for length, HC, and MUAC. There were no obvious signs of digit preference for weight measurement. The distribution of terminal digits post-training was evenly distributed for all measures. Similar patterns exist when examining the DPS. The DPS for length, HC and MUAC prior to the training exceeded the acceptable limit, while the DPS post-training were below the acceptable cutoff of 20.

### Evaluation of quality- means and standard deviations of anthropometric indices

Table 3 summarizes the means and standard deviations for length-for-age (LAZ), weight-for-age (WAZ), and weight-for-length (WLZ), expressed as z scores. There was a substantial loss in sample size when examining WLZ using WHO growth standards with 12% data loss (n = 9) in the pre- and 22% loss (n = 17) in the post-training group. Except for WLZ of children <1 month of age, the standard deviations of all indices exceeded acceptable values both pre- and post-training. There were no differences in WAZ and WLZ pre- and post-training, but there was a statistically significant increase in LAZ post-training (p<0.01). There were no significant

**Table 1. Sample characteristics among pre- and post-intervention groups, Manyatta, Kenya.**

| | Pre-intervention, n = 75 | Post-intervention, n = 76 | p-value[4] |
|---|---|---|---|
| **Age category, n (%)** | | | |
| <1 day | 15 (20.0) | 21 (27.6) | 0.4821 |
| 1 day– 5 months | 28 (37.3) | 20 (26.3) | |
| 6–23 months | 23 (30.7) | 25 (32.89) | |
| 24–59 months | 9 (12.0) | 10 (13.1) | |
| **Sex, n (%)** | | | |
| Female | 31 (41.3) | 35 (46.1) | 0.5589 |
| **Anthropometric measurements, mean (SD)** | | | |
| Weight, kg | 5.0 (3.8) | 4.8 (3.5) | 0.7543 |
| Length, cm | 62.0 (18.0) | 60.0 (17.6) | 0.4899 |
| Head circumference (HC), cm | 39.0 (6.9) | 37.9 (7.4) | 0.3509 |
| Mid-Upper Arm Circumference (MUAC), cm | 11.0 (3.0) | 10.2 (3.0) | 0.1064 |
| **Nutritional status, n (%)** | | | |
| Stunting (LAZ[1]<-2SD) | 24 (32.0) | 38 (50.0) | 0.0246 |
| Wasting (WLZ[2]<-2SD) | 58 (77.3) | 54 (71.2) | 0.3780 |
| Underweight (WAZ[3]<-2) | 40 (53.3) | 50 (65.8) | 0.1188 |

[1] LAZ: Length-for-age z-score

[2] WLZ: Length-for-weight z-score

[3] WAZ: Weight-for-age z-score

[4] p-values calculated using Chi Sq tests (age, sex, nutritional status) or t-tests (anthropometric measurements)

**Table 2. Manual anthropometry digit preference scores[1] pre- and post-intervention, Manyatta, Kenya.**

| | Pre- intervention, (N = 75) n(%) | | | | Post- intervention, (N = 76) n(%) | | | |
|---|---|---|---|---|---|---|---|---|
| | Length | Weight | HC | MUAC | Length | Weight | HC | MUAC |
| **0.0** | 65 (86.7) | 15 (20.0) | 57 (77.3) | 54 (72.0) | 3 (4.0) | 10 (13.2) | 5 (6.6) | 2 (2.6) |
| **0.1** | - | 2 (2.7) | - | - | 12 (15.6) | 8 (10.5) | 11 (14.5) | 18 (23.7) |
| **0.2** | - | 6 (8.0) | - | - | 7 (9.2) | 9 (11.8) | 9 (11.8) | 10 (13.2) |
| **0.3** | - | 9 (12.0) | - | - | 13 (17.1) | 4 (5.3) | 3 (3.9) | 9 (11.8) |
| **0.4** | - | 6 (8.0) | - | - | 5 (6.6) | 9 (11.8) | 9 (11.8) | 5 (6.6) |
| **0.5** | 10 (13.3) | 6 (8.0) | 17 (22.7) | 21 (28.0) | 6 (7.9) | 9 (11.8) | 8 (10.5) | 9 (11.8) |
| **0.6** | - | 11 (14.7) | - | - | 7 (9.2) | 10 (13.2) | 13 (17.1) | 5 (6.6) |
| **0.7** | - | 9 (12.0) | - | - | 8 (10.5) | 4 (5.3) | 1 (1.3) | 5 (6.6) |
| **0.8** | - | 5 (6.7) | - | - | 8 (10.5) | 6 (7.9) | 12 (15.8) | 8 (10.5) |
| **0.9** | - | 6 (8.0) | - | - | 7 (9.2) | 7 (9.2) | 5 (6.6) | 5 (6.6) |
| **Digit preference score [1]** | 86.2 | 15.3 | 78.1 | 74.3 | 10.4 | 9.5 | 16.6 | 18.4 |

[1] Digit preference scores computed using Mark Myatt and Ernest Guevarra (2022).

nipnTK: National Information Platforms for Nutrition

Anthropometric Data Toolkit. https://nutriverse.io/nipnTK/,

https://github.com/nutriverse/nipnTK

DPS<20 is acceptable; ≥20 indicates digit preference is problematic

**Table 3. Means and standard deviations for manual anthropometric indices, Manyatta, Kenya.**

|  | Pre-training, (N = 75) | | Post-training, (N = 76) | | p-value[1] | Expected SD for high data quality [27] |
|---|---|---|---|---|---|---|
|  | n | Mean (SD) | n | Mean (SD) |  |  |
| LAZ[2] overall | 75 | -1.1 (2.6) | 76 | -2.5 (2.9) | 0.0018 | **1.1–1.3** |
| < 1 months | 35 | -0.8 (2.8) | 31 | -3.0 (3.2) |  |  |
| 1–59 months | 40 | -1.4 (2.3) | 45 | -2.2 (2.7) |  |  |
| WAZ[3] overall | 75 | -2.6 (2.3) | 76 | -3.2 (2.4) | 0.0962 | **1.0–1.2** |
| < 1 months | 35 | -2.0 (2.2) | 31 | -2.9 (2.2) |  |  |
| 1–59 m months | 40 | -3.1 (2.3) | 45 | -3.5 (2.5) |  |  |
| WLZ[4] overall | 66 | -3.1 (1.8) | 59 | -2.9 (2.2) | 0.4777 | **0.85–1.1** |
| < 1 months | 28 | -2.6 (1.1) | 15 | -1.5 (1.3) |  |  |
| 1–59 months | 38 | -3.5 (2.1) | 44 | -3.3 (2.3) |  |  |

[1] p-values comparing overall pre- and post-training mean z-scores calculated using t-tests

[2] LAZ: Length-for-age z-score

[3] WLZ: Length-for-weight z-score

[4] WAZ: Weight-for-age z-score

changes between the SDs for LAZ and WAZ pre- and post-training overall, and when stratified by age (<1 month vs 1–59 months as well as <6 months vs 6–59 months).

## Evaluation of precision-technical errors of measurement

Table 4 presents the TEMs and rTEMs specific to the post-training measures.

The TEMs for length, weight, HC, and MUAC were, 0.32, 0.01, 0.18, and 0.13 respectively. The rTEMs for length, weight, HC, and MUAC were 0.53%, 0.29%, 0.48%, and 1.24%, respectively. All TEMs and rTEMs were within the acceptable range.

**Table 4. Manual anthropometry technical errors of measurement for post-intervention measures, Manyatta, Kenya.**

|  | Length (cm) | Weight (kg) | Mid-Upper Arm Circumference (cm) | Head Circumference (cm) |
|---|---|---|---|---|
| TEM[A] | 0.32 | 0.01 | 0.13 | 0.18 |
| Acceptable TEM [32] | 0.35 | 0.17 | 0.26 | - |
| VAV | 60.00 | 4.84 | 10.22 | 37.88 |
| Relative TEM (% TEM)[c] | 0.53% | 0.29% | 1.24% | 0.48% |

The technical error of measurement (TEM) is defined as the standard deviation of differences between repeated measures in the unit of the measurement, using the following equation

[A] Equation 1: absolute technical errors of measurement (TEM) $= \sqrt{\frac{\Sigma d_i^2}{2n}}$

Where:

$\Sigma d_i^2$ = Squared summation of deviations, n = number of individuals measured, and i = number of deviations

[C] Equation 2: relative TEM $= 100 \; x \; \frac{TEM}{VAV}$

Where TEM = technical error of measurement expressed as %, VAV = variable average value, the relative TEM (%TEM), and the coefficient of reliability (R) were the statistical tests used to assess intra- and inter-observer reliability. The TEM was defined as the standard deviation of differences between repeated measures in the unit of the measurement (e.g., TEM for height measured in centimeters is cm), using the following equation:

Skillful anthropometrists relative technical errors of measurement (%TEM) cutoff $\leq 1.5\%$ [25]

## Accuracy- spearman correlation and Bland Altman Plots

Spearman correlation coefficients (Fig 1) comparing the manual measures to the 3D scans for length, MUAC, and HC were 0.99, 0.91, and 0.93, respectively. While the manual measures were highly correlated with the scans, the mean differences between scans and manual measures for length, MUAC, and HC were 1.61 cm, -0.20 cm, and 2.27 cm, respectively. These results suggest that the scans overestimate length by 1.61 cm, underestimate MUAC by 0.20 cm, and overestimate HC by 2.27 cm.

While there were challenges in securing data at the CHOA site, findings were complementary to those in the Kenya site. Among the 3 cases, standard anthropometry measurements were feasible and showed high precision (rTEMs for manual length, MUAC, and HC were 0.62%, 0.96%, and 1.80% respectively). For 3D scans, precision for duplicate scans was within acceptable limits when measuring length (rTEM = 1.05%), but the software had more difficulty capturing precise measurements for MUAC (rTEM = 4.71%) and HC (rTEM = 1.62%).

## Qualitative findings

The post-intervention survey revealed that all participants felt that training in manual anthropometry improved the accuracy of their measurements. Additionally, all participants reported feeling confident in their ability to perform manual anthropometry. While most participants (66.7%) believed that 3D imaging reduced measurement time in comparison to manual anthropometry, all participants overall preferred the use of manual anthropometry.

The qualitative findings from the in-depth interviews revealed that the team had a clear preference for manual anthropometry over the 3D imaging software as they felt the 3D imaging software required more time, better lighting, improved morgue environment, and training to ensure an accurate scan.

*"We would take manual anthropometric measurements more seriously and would choose it well over 3D scanning. . .A lot of movement and manipulation of the camera to capture the entire body. And many times for 3D imaging, you have to repeat the process over and over and over again for you to be able to get the entire body into the screen. So it takes quite a bit more time. . .The boards work really well for us. It's a stable board. . . it's something we opt for over any other methods."*

Additionally, study investigators cited challenges in using the software when lighting was insufficient or when morgue environments varied.

*"For what we experienced on the 3D, we had a few issues . . . our autopsy table had a fixed length and was not adjustable, so it was hard to get the complete image as you scan. Many times, we had issues with lighting systems. This made us end up with cut images—images with some parts of the body missing. So that called for checking and re-checking of images for quite a long period of time."*

Lastly, study investigators noted postmortem-specific challenges to manual anthropometry and understood the implications of taking careful measurement and attention to details to ensure data quality and minimize measurement.

*"With rigor mortis, you will find that children stiffening, even the legs stiffening in some specific direction. If you are not able to manipulate them properly, one will end up with increased length as opposed to getting the accurate length. So that also required a lot of keenness."*

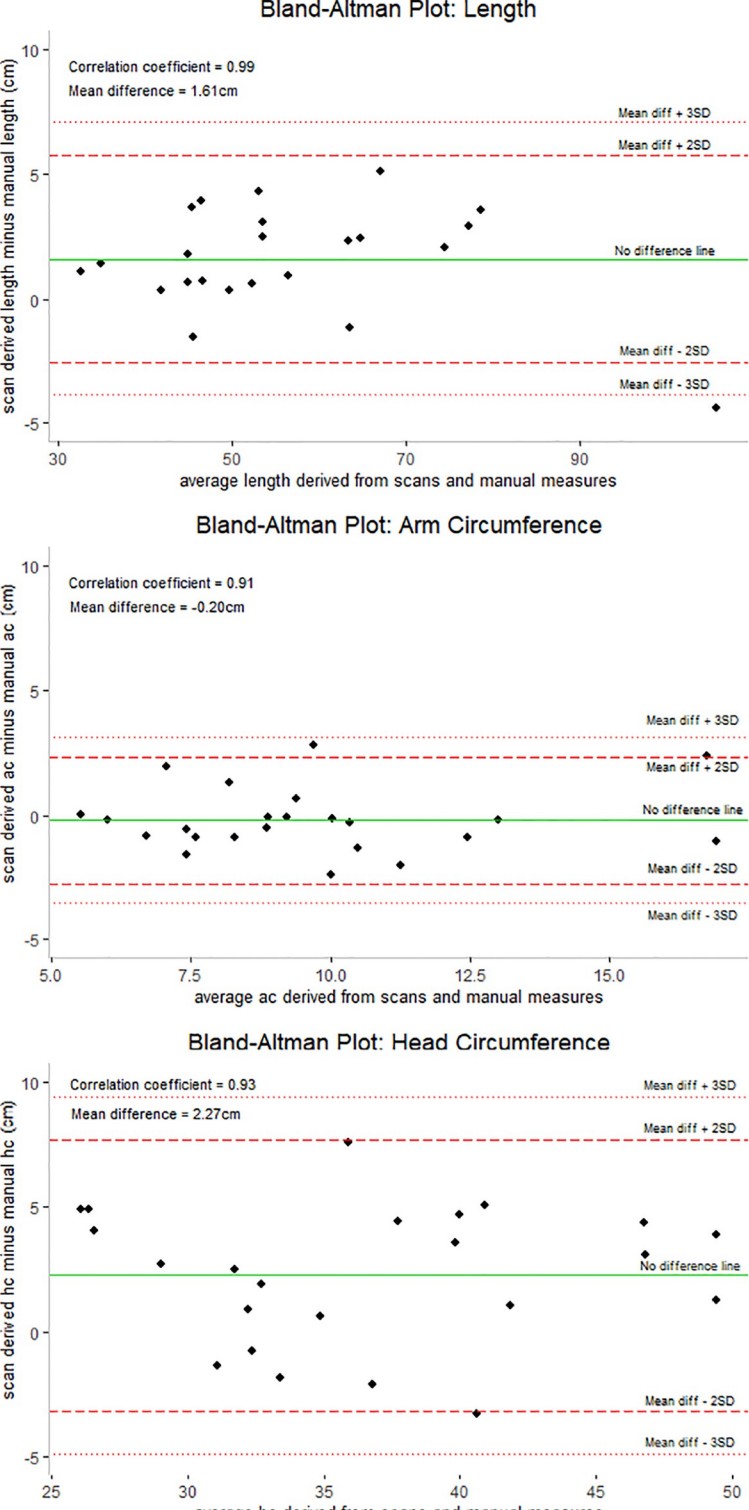

**Fig 1. Bland Altman Plots comparing manual anthropometry and 3D imaging, Manyatta, Kenya. Y-axis**: the difference between the scan measurements and manual measurements; **X-axis**: the average of the scan and manual measures; **Dotted lines**: represent the mean difference ± 3 standard deviations; **Dashed lines**: represent the mean difference ± 2 SD; **Solid line**: across the plot is the no difference line. Black points on the chart represent the 23 cases for which we had viable 3D scan data. Spearman correlation coefficients were examined to measure the strength of the relationship between scans and manual measures. AC: Arm Circumference, HC: Head Circumference.

> *"The challenge in checking MUAC with tape measure comes when the subject you are measuring has reduced skin turgor. That is the skin of the arm becomes floppy. So that one might give you a lesser MUAC."*

## Discussion

Following training on manual anthropometry and use of standard equipment for post-mortem assessment of nutritional status, data quality and precision improved; however, standard deviations of anthropometric indices pre- and post-training exceeded acceptable values. 3D imaging scans overestimated length by approximately 1.6 cm, underestimated MUAC by 0.2 cm, and overestimated HC by 2.3 cm. The presence of rigor mortis did not impede the collection or quality of manual anthropometry measurements; however, additional care and pressure are critical to ensuring high quality data.

Digit preference improved for length, HC and MUAC following the training. There was no evidence of digit preference for weight pre- or post-training, which is likely due to how the measurements were taken. Weight was read from a digital scale, while length and circumference measurements were reliant on the anthropometrist's ability to read a tape measure accurately. Previous studies among living children have shown that the SD of anthropometric z-scores are reasonably consistent across populations, irrespective of nutritional status, and thus can be used to assess the quality of anthropometric data [27]. The SD for all anthropometric indices exceeded acceptable limits both pre- and post-training, and sensitivity analyses revealed that high SDs for LAZ and WAZ were unlikely to be explained by age. If we continue with the conclusion that the intervention may have improved data quality and precision, then the persistently high SDs may be explained by capturing anthropometric measurements of small, severely ill children.

We also noted a decrease in sample size when examining WLZ scores. This is because nearly one-fourth of children in this sample fell below 45 cm, or the smallest length captured by the WHO growth standards when calculating WLZ [30]. The WHO growth standards were based on a healthy population of children, receiving optimal nutrition, raised in optimal environments, and receiving optimal healthcare—unlike the cases captured in CHAMPS. Many of the CHAMPS cases, at the end of life, had severe malnutrition and had body sizes not compatible with postnatal life and survival based on their chronologic age. Future research might consider application of the INTERGROWTH-21 (IG21-GS) standards [31] to classify nutritional status of children that fall outside of the WHO growth standards, such as in the case of severely ill cohorts of young children in CHAMPS.

This study has multiple strengths. First, to our knowledge, no research has been conducted on the feasibility of using gold-standard anthropometric assessment in the postmortem setting. Assessment of malnutrition and standardization of growth within the field of nutrition is typically based on z-scores derived from the 2006 WHO's Multicentre Growth Reference Study (MGRS). These standards are based on healthy, living children. Utilizing anthropometric data from CHAMPS, a large, multi-site surveillance system designed to elucidate the causes of U5M in high mortality regions of the world, may help inform the possible ranges of anthropometric deficits in severely ill populations. Second, our project captured staff reflections of conducting manual anthropometry of young children in field-based and clinical-morgue postmortem settings. These qualitative findings may prove useful in informing strategies to improve the accuracy of post-mortem anthropometry.

This project was also subject to several limitations. First, in the CHOA site, we encountered unexpected obstacles in reaching our goal sample size due to limited time to perform the

manual and 3D imaging anthropometric measurements before autopsies were performed. Further, the added data collection steps placed a significant burden on clinical staff and led to disruption of their workflow. Second, in Kenya, challenges arose with the 3D imaging software. The software settings were subject to user error and were altered during data collection, which resulted in a compromised final sample size. Among the viable scans, our results suggest that the scans overestimated both length and HC. These findings are aligned with a recent study [24] and further suggest that before 3D imaging can be considered a viable, accurate alternative to manual anthropometry, adjustment of the technology and additional user testing is warranted to ensure reliable anthropometric measures.

## Conclusions

Collection of quality anthropometric data following implementation of standardized training and equipment is feasible and reliable in postmortem field studies. While 3D imaging may be an accurate alternative to manual anthropometry, technology adjustments are needed to ensure accuracy and usability. Future research on the appropriate use of standards to define malnutrition among severely ill populations, including those in the post-mortem setting, are needed to elucidate our understanding of the role of malnutrition in U5M.

## Supporting information

**S1 Data.**
(XLSX)

**S2 Data.**
(XLSX)

## Acknowledgments

The Child Health and Mortality Prevention Surveillance network would like to extend sincere appreciation to all the families who participated. Additionally, special thanks to Afrin Jahan for her analysis replication and figure development of Child Health and Mortality Prevention Surveillance (CHAMPS) network data related to this work.

## Author Contributions

**Conceptualization:** Priya M. Gupta, Kasthuri Sivalogan, Richard Oliech, Jamie Klein, Victor Akelo, Parminder S. Suchdev.

**Data curation:** Priya M. Gupta, Richard Oliech, Eugene Alexander, Dickson Gethi, Victor Akelo, Parminder S. Suchdev.

**Formal analysis:** Priya M. Gupta, O. Yaw. Addo.

**Investigation:** Priya M. Gupta, Eugene Alexander.

**Methodology:** Priya M. Gupta, Eugene Alexander, O. Yaw. Addo, Parminder S. Suchdev.

**Resources:** Priya M. Gupta.

**Software:** Priya M. Gupta, Eugene Alexander.

**Supervision:** Kasthuri Sivalogan, Eugene Alexander, O. Yaw. Addo, Parminder S. Suchdev.

**Validation:** Priya M. Gupta.

**Visualization:** Priya M. Gupta.

**Writing – original draft:** Priya M. Gupta.

**Writing – review & editing:** Priya M. Gupta, Kasthuri Sivalogan, Richard Oliech, Eugene Alexander, Jamie Klein, O. Yaw. Addo, Dickson Gethi, Victor Akelo, Dianna M. Blau, Parminder S. Suchdev.

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
