## [Decision Letter · Decision Letter 0]

19 Apr 2023

PONE-D-23-01635Feasibility and accuracy of performing manual anthropometry in the postmortem settingPLOS ONE

Dear Dr. Gupta,

Thank you for submitting your manuscript to PLOS ONE. After careful consideration, we feel that it has merit but does not fully meet PLOS ONE’s publication criteria as it currently stands. Therefore, we invite you to submit a revised version of the manuscript that addresses the points raised during the review process. Your manuscript has been assessed by two independent reviewers and we ask you to incorporate their recommendations.

Please submit your revised manuscript by Jun 03 2023 11:59PM.  If you will need more time than this to complete your revisions, please reply to this message or contact the journal office at plosone@plos.org. Please include the following items when submitting your revised manuscript:A rebuttal letter that responds to each point raised by the academic editor and reviewer(s). You should upload this letter as a separate file labeled 'Response to Reviewers'.A marked-up copy of your manuscript that highlights changes made to the original version. You should upload this as a separate file labeled 'Revised Manuscript with Track Changes'.An unmarked version of your revised paper without tracked changes. You should upload this as a separate file labeled 'Manuscript'.

We look forward to receiving your revised manuscript.

Kind regards,

Olutosin Ademola Otekunrin, PhD

Academic Editor

PLOS ONE

Journal Requirements:

2. Please include a complete copy of PLOS’ questionnaire on inclusivity in global research in your revised manuscript. Our policy for research in this area aims to improve transparency in the reporting of research performed outside of researchers’ own country or community. The policy applies to researchers who have travelled to a different country to conduct research, research with Indigenous populations or their lands, and research on cultural artefacts. The questionnaire can also be requested at the journal’s discretion for any other submissions, even if these conditions are not met.  

Please find more information on the policy and a link to download a blank copy of the questionnaire here: https://journals.plos.org/plosone/s/best-practices-in-research-reporting. 

Please upload a completed version of your questionnaire as Supporting Information when you resubmit your manuscript.

3.Please provide additional details regarding participant consent. In the ethics statement in the Methods and online submission information, please ensure that you have specified (1) whether consent was informed and (2) what type you obtained (for instance, written or verbal, and if verbal, how it was documented and witnessed). If your study included minors, state whether you obtained consent from parents or guardians. If the need for consent was waived by the ethics committee, please include this information.

"I have read the journal's policy and the authors of this manuscript have the following competing interests:  Eugene Alexander holds an ownership position in Body Surface Translations and therefore has a financial interest in the success of the 3D testing device described in this study. Data were blinded and not shared with Mr. Alexander until completion of draft manuscript.

Additional disclosure: The findings and conclusions in this report are those of the authors and do not necessarily represent the official position of the Centers for Disease Control and Prevention."

Please confirm that this does not alter your adherence to all PLOS ONE policies on sharing data and materials, by including the following statement: ""This does not alter our adherence to  PLOS ONE policies on sharing data and materials.” (as detailed online in our guide for authors http://journals.plos.org/plosone/s/competing-interests).  

If there are restrictions on sharing of data and/or materials, please state these. Please note that we cannot proceed with consideration of your article until this information has been declared. 

7. We note that you have included the phrase “data not shown” in your manuscript. Unfortunately, this does not meet our data sharing requirements. PLOS does not permit references to inaccessible data. We require that authors provide all relevant data within the paper, Supporting Information files, or in an acceptable, public repository. Please add a citation to support this phrase or upload the data that corresponds with these findings to a stable repository (such as Figshare or Dryad) and provide and URLs, DOIs, or accession numbers that may be used to access these data. Or, if the data are not a core part of the research being presented in your study, we ask that you remove the phrase that refers to these data.

8. Please include a separate caption for figure in your manuscript.

Reviewers' comments:

Reviewer's Responses to Questions

**Comments to the Author**

1. Is the manuscript technically sound, and do the data support the conclusions?

Reviewer #1: No

Reviewer #2: Yes

2. Has the statistical analysis been performed appropriately and rigorously? 

Reviewer #1: No

Reviewer #2: Yes

3. Have the authors made all data underlying the findings in their manuscript fully available?

Reviewer #1: No

Reviewer #2: No

4. Is the manuscript presented in an intelligible fashion and written in standard English?

Reviewer #1: Yes

Reviewer #2: Yes

5. Review Comments to the Author

Reviewer #1: The study is a very interesting study that will provide informaiton on how the errors in manual anthropometry can be improved in a post-mortem setting. Information claimed to be evaluated in the study will be very vital in nutrition assessment of children in post-mortem settings. However, the following observations were made during review:

1. The title of the article seem not to be approriate. Suggested title is presented in the comment section.

2. comments on the abstract, introduction and materials and methods, and results were made in the manuscript.

3. Generally, authors seem not to define the objectives of the study properly and this is reflecting in the methodology, and result sections.

4. Authors should interprete what is on the table correctly in the result section.

5. The authors need to state the objectives of the study clearly and present results based on these objectives.

6. Statistical analysis carried out was not clearly stated in the methodology. It is not appropriate to have to look at the table before having an idea of the statistics carried out.

7. Figures indicated on the manuscript were not seen.

8. In the methodology, authors claimed to do a survey to collect information on whether the participants believed training on manual anthropometry improved the accuracy of the measurements, whether 3D imaging reduced the time to measure, and asked about the participants preference in measuring using manual anthropometry or the 3D imaging technology. In addition, authors also claimed to conducted a 60-minute in-depth interview with the single lead site technician to collect qualitative feedback on the team’s experience with performing manual anthropometry and ease of using the 3D imaging software. The results for these survey and qualitative study are not clear in the result sections as well as the tables.

Results were presented not indicating whether it is for manual anthropometry or 3D imaging (Tables 1-4). Although, I suppose that is for manual anthropometry. The results for pre- and post-training for the 3D scan were never presented and figure 1 was comparing manual anthropometry with 3 D imaging (although the figures were not seen).

Results on qualitative feedbacks were only presented for 3D imaging and not manual anthropometry.

9. Results on whether participants believed training on manual anthropometry improved the accuracy of the measurement or 3D imaging reduced the time to measure were not presented at all. Also, participants preference in measuring using manual anthropometry or 3D imaging technology was not presented in the result section.

10. Discussion section needs to be re-written to reflect exactly what is in the results. In addition, results need to be properly discussed in line with findings from previous studies and implications should be discussed clearly and appropriately. Assumptions in the result sections is not approriate.

11.Authors did not have conclusion section at all.

12. There is need for proper organization of the content of the manuscripts for coherence.

13. Some sentences seem complicated and difficult to understand. The authors are advised to seek for professional English editing service to check the revised manuscript for grammar, syntax and style errors.

Reviewer #2: PONE-D-23-01635

Feasibility and accuracy of performing manual anthropometry in the post-mortem setting

Comments to the editor and authors

This is an important topic because the findings add to the existing evidence on causes of death due to anthropometric deficits in children. This study is novel and a useful contribution to the body of evidence on child health and nutrition. The paper fits the PlosOne journal’s aim and will be interesting to your readership.

The analysis is comprehensive and accurate. Limitations and strengths of the analysis have been declared adequately. Data analysis and results were adequately done and well presented. This paper deserves to be published.

Please, find below suggested minor comments and suggestions for your consideration to further improve your manuscript:

Comments:

Title

1. The current title of the article should be modified to reflect manual anthropometry as well as 3D imaging and the target or study population - children under 5 years.

Abstract

1. Please, could the conclusion “Future research on the appropriate use of current growth standards to define malnutrition in this severely ill population is needed” be revised to reflect the topic of interest. …. This severely ill population is not very clear, I thought the study setting was post-mortem.

Method

1. In paragraph 3 …. weight, and circumference measurements using two…. please could indicate which circumference measurement you are referring to?

2. Please revise the sentence in paragraph 3 …. ‘Following the training, two unique site staff each performed manual anthropometry on 76 new cases, for a total of 2 manual measures per case’ and make it simpler and clearer.

3. Authors should check if the sentence in paragraph 3 “Following data collection, it was found that the software… settings had been inadvertently altered on the scanner resulting in viable scan data on only 23 cases.” is communicating the right message, because if the software was inadvertently altered then it could me the data was not viable. I may be wrong. If that was true, then how did it impact on the findings?

4. Clarity needed what actually happened? ……‘Manual anthropometry was to be performed prior to the start of the diagnostic autopsy. Significant challenges arose during data collection which resulted in a limited sample size of 3 cases; thus, our results will focus on the Kenya site’.

5. What was the duration of the training? What was the duration of the data collection?

6. Please indicate how you analysed the qualitative data, and how you utilized the data.

Results

1. ‘There was a substantial loss in sample size when examining WLZ using WHO growth standards with 12% data loss in the pre- and 22% loss in the post-training group’.

How was LAZ also affected given that there was data lost for WLZ?

2. Why will authors talk about results that are not available?

‘While there were challenges in securing data at the CHOA site, findings were complementary to those in the Kenya site (data not shown)’.

Discussion

1. Authors, please explain why rigor mortis will not impede manual anthropometry measurements, this is because the qualitative findings show that it could be a challenge to get accurate measurements.

2. Table 3 in the results section and Paragraph 3 in the discussion have some repetitions, this happened because you cited literature in your results section. Authors should consider to present only results under the results section.

3. Check WHO-GS should be written as WHO-MGRS.

4. ‘Future research on the appropriate use of standards to define malnutrition among severely ill populations will elucidate our understanding of the role of malnutrition in U5M and inform future malnutrition-specific U5M reduction interventions’ I think I know what you are trying to say but I am wandering if this appropriate recommendation because you did not work with severely sick children. Please consider to revise your recommendation.

General comments

1. Please number the lines for the sentences in your manuscript. It helps reviewers to give feedback easily.

6. PLOS authors have the option to publish the peer review history of their article (what does this mean?). If published, this will include your full peer review and any attached files.

Reviewer #1: **Yes: **Oluwafunke Akinbule

Reviewer #2: No

---

## [Author Response · Author response to Decision Letter 0]

22 Jun 2023

Full response to editor and reviewer comments has been uploaded.

---

## [Decision Letter · Decision Letter 1]

15 Aug 2023

PONE-D-23-01635R1

Impact of anthropometry training and feasibility of 3D imaging on anthropometry data quality among children under five years in a postmortem setting

Dear Dr. Suchdev,

Thank you for submitting your manuscript to PLOS ONE. After careful consideration, we feel that it has merit but does not fully meet PLOS ONE’s publication criteria as it currently stands. Therefore, we invite you to submit a revised version of the manuscript that addresses the points raised during the review process.

ACADEMIC EDITOR:

Please, kindly attend to the review comments/suggestions of the two reviewers attached to this email.

We look forward to receiving your revised manuscript.

Kind regards,

Olutosin Ademola Otekunrin

Academic Editor

PLOS ONE

Journal Requirements:

Reviewers' comments:

Reviewer's Responses to Questions

**Comments to the Author**

1. If the authors have adequately addressed your comments raised in a previous round of review and you feel that this manuscript is now acceptable for publication, you may indicate that here to bypass the “Comments to the Author” section, enter your conflict of interest statement in the “Confidential to Editor” section, and submit your "Accept" recommendation.

Reviewer #1: (No Response)

Reviewer #2: All comments have been addressed

2. Is the manuscript technically sound, and do the data support the conclusions?

Reviewer #1: Yes

Reviewer #2: Yes

3. Has the statistical analysis been performed appropriately and rigorously? 

Reviewer #1: Yes

Reviewer #2: N/A

4. Have the authors made all data underlying the findings in their manuscript fully available?

Reviewer #1: No

Reviewer #2: Yes

5. Is the manuscript presented in an intelligible fashion and written in standard English?

Reviewer #1: Yes

Reviewer #2: Yes

6. Review Comments to the Author

Reviewer #1: Authors did not attend to some of the comments raised earlier. These needs to be addressed before acceptance.

Reviewer #2: I wish to thank the authors for considering my comments in the revision of this manuscript, however, I have one comment on how qualitative data was analysed. I have seen your response below. I wish you were more elaborate on how you analysed the qualitative data. I am concern because most of us researchers don’t pay keen attention to qualitative data especially when we are using both quantitative and qualitative methods in a study.

If the academic editor is satisfied with your response, then PLOS-One can go ahead with publication of the manuscript.

Thank you once again for the opportunity to review your paper.

7. PLOS authors have the option to publish the peer review history of their article (what does this mean?). If published, this will include your full peer review and any attached files.

Reviewer #1: No

Reviewer #2: **Yes: **Gloria Odei Obeng-Amoako

---

## [Author Response · Author response to Decision Letter 1]

25 Aug 2023

Thank you for the thoughtful review of our revised manuscript. We have carefully revised and responded to each point raised by the reviewers noted below in red font. We have also submitted a revised manuscript in track changes, as well as an unmarked revised manuscript.

Reviewer #1

The research design is not clearly stated. You need to state the design for the quantitative and the qualitative study. What design was used for the pre- and post- conducted. This is a very important.

- We have revised the first sentence of the Methods section to further clarify the research design, emphasizing the both quantitative and qualitative research approach was used.

“This longitudinal quality improvement study adopted a mixed-methods approach utilizing quantitative and qualitative research on the experience conducting manual anthropometry and 3D imaging in the postmortem setting.”

Kindly give a detailed information on the content of the interview.

- We believe we have adequately described the content of the interview on lines 205-206 as stated “…to collect qualitative feedback on the team’s experience with performing manual anthropometry and ease of using the 3D imaging software.”

The title of this result should be stated before the interpretation.

- We are unclear about this comment. Hopefully it is resolved with response to other comments about interpretation of Table 1 below.

It is better to report the improvement or otherwise that happened post-intervention.

- We have changed the order and wording in lines 226-231

There is significant differences in the sample characteristics for stunting (as shown in table 1). This should reflect in the result interpretation and the p-value should be stated. The type of statistics carried out should be clearly stated in the methodology.

- We have reworded the interpretation of findings in Table 1. The statistical approach is described in the Methods--“Pearson’s Chi-Square tests (categorical variables) or t-tests (continuous variables) were used to evaluate differences between pre-intervention and post-interventions groups-- and also in the legend of the table.

Digital Preference Score of the respondents anthropometric indices is suggested as title.

Digital preference score was used to assess respondents' anthropometric indices and that should reflect in the title

- We support keeping the subheading the same as we report both absolute digit values and calculated DPS in table 2. To further clarify, DPS are not related to calculation of anthropometric indices and simply reflect interpretations of the raw recorded anthropometric measurements.

Mean anthropomentry of the respondents. This result should come before the DPS

- As noted above, the DPS is calculated before entering the measurements into statistical software for calculation of anthropometric indices. This we support keeping table 3 after table 2. 

Except the WLZ of the children of ages <1 month which was within the acceptable SD for high quality data.

- We appreciate this comment and have modified the text accordingly.

State the p value (e.g. at p<0.01) for where there is statistical difference.

- We have added the p-value noted in the table.

This statement is not correct. Kindly check your table clearly and ensure that what is on the table is in agreement with what is interpreted here in the result section.

This statement is not correct. Kindly check your table clearly and ensure that what is on the table is in agreement with what is interpreted here in the result section. There was rather increase in the WLZ

- We have moved up this sentence to avoid confusion and also added the sample size of number of children who dropped out when calculated WLZ (n=9 and n=17 in the pre- and post-training groups, respectively. The calculations of data loss are correct.

You cannot assume that it is due to how the measurements were taken. You need to discuss this in line with previous findings.

- Thanks for this comment. We explain the rationale for this statement in the following sentence. “Weight was read from a digital scale, while length and circumference measurements were reliant on the anthropometrist’s ability to read a tape measure accurately.”

Reviewer #2

I wish to thank the authors for considering my comments in the revision of this manuscript, however, I have one comment on how qualitative data was analysed. I have seen your response below. I wish you were more elaborate on how you analysed the qualitative data. I am concern because most of us researchers don’t pay keen attention to qualitative data especially when we are using both quantitative and qualitative methods in a study.

- We appreciate this request for additional description of the qualitative analysis and have revised the description as follows (line 210-212): “The qualitative data were analyzed using simple frequencies and applying manual thematic analysis; findings informed the implementation of manual anthropometric measurements across the CHAMPS Network.”

---

## [Decision Letter · Decision Letter 2]

12 Sep 2023

Impact of anthropometry training and feasibility of 3D imaging on anthropometry data quality among children under five years in a postmortem setting PONE-D-23-01635R2Dear Dr. Suchdev,We are pleased to inform you that your manuscript has been judged scientifically suitable for publication and will be formally accepted for publication once it meets all outstanding technical requirements. Within one week, you will receive an e-mail detailing the required amendments. When these have been addressed, you will receive a formal acceptance letter and your manuscript will be scheduled for publication. An invoice for payment will follow shortly after the formal acceptance. To ensure an efficient process, please log into Editorial Manager at http://www.editorialmanager.com/pone/, click the Update My Information; link at the top of the page, and double check that your user information is up-to-date. If you have any billing related questions, please contact our Author Billing department directly at authorbilling@plos.org. If your institution or institutions have a press office, please notify them about your upcoming paper to help maximize its impact. If they will be preparing press materials, please inform our press team as soon as possible -- no later than 48 hours after receiving the formal acceptance. Your manuscript will remain under strict press embargo until 2 pm Eastern Time on the date of publication. For more information, please contact onepress@plos.org.Kind regards,Olutosin Ademola Otekunrin, PhDAcademic Editor PLOS ONE Additional Editor Comments (optional):Please, kindly modify the title as "Impact of anthropometry training and feasibility of 3D imaging on anthropometry data quality among under-five children in a postmortem setting"Reviewers comments: Reviewer's Responses to Questions

**Comments to the Author**

1. If the authors have adequately addressed your comments raised in a previous round of review and you feel that this manuscript is now acceptable for publication, you may indicate that here to bypass the “Comments to the Author” section, enter your conflict of interest statement in the “Confidential to Editor” section, and submit your "Accept" recommendation.

Reviewer #1: All comments have been addressed

Reviewer #2: All comments have been addressed

2. Is the manuscript technically sound, and do the data support the conclusions?

Reviewer #1: Yes

Reviewer #2: Yes

3. Has the statistical analysis been performed appropriately and rigorously? 

Reviewer #1: Yes

Reviewer #2: Yes

4. Have the authors made all data underlying the findings in their manuscript fully available?

Reviewer #1: Yes

Reviewer #2: Yes

5. Is the manuscript presented in an intelligible fashion and written in standard English?

Reviewer #1: Yes

Reviewer #2: Yes

6. Review Comments to the Author

Reviewer #1: (No Response)

Reviewer #2: Authors have addressed all the comments. Data analysis method for the qualitative data has been described.

Thank you

7. PLOS authors have the option to publish the peer review history of their article (what does this mean?). If published, this will include your full peer review and any attached files.

Reviewer #1: No

Reviewer #2: No

---

## [Editor Report · Acceptance letter]

19 Sep 2023

PONE-D-23-01635R2 

Impact of anthropometry training and feasibility of 3D imaging on anthropometry data quality among children under five years in a postmortem setting 

Dear Dr. Suchdev:

I'm pleased to inform you that your manuscript has been deemed suitable for publication in PLOS ONE. Congratulations! Your manuscript is now with our production department. 

Kind regards, 

on behalf of

Dr. Olutosin Ademola Otekunrin 

Academic Editor

PLOS ONE